# Real-time 3D tracking of swimming microbes using digital holographic microscopy and deep learning

Samuel A. Matthews[1], Carlos Coelho[1], Erick E. Rodriguez Salas[1], Emma E. Brock[1], Victoria J. Hodge[2], James A. Walker[2], Laurence G. Wilson[1]*

1 School of Physics, Engineering and Technology, University of York, Heslington, York, United Kingdom,
2 Department of Computer Science, Deramore Lane, York, United Kingdom

* laurence.wilson@york.ac.uk

## Abstract

The three-dimensional swimming tracks of motile microorganisms can be used to identify their species, which holds promise for the rapid identification of bacterial pathogens. The tracks also provide detailed information on the cells' responses to external stimuli such as chemical gradients and physical objects. Digital holographic microscopy (DHM) is a well-established, but computationally intensive method for obtaining three-dimensional cell tracks from video microscopy data. We demonstrate that a common neural network (NN) accelerates the analysis of holographic data by an order of magnitude, enabling its use on single-board computers and in real time. We establish a heuristic relationship between the distance of a cell from the focal plane and the size of the bounding box assigned to it by the NN, allowing us to rapidly localise cells in three dimensions as they swim. This technique opens the possibility of providing real-time feedback in experiments, for example by monitoring and adapting the supply of nutrients to a microbial bioreactor in response to changes in the swimming phenotype of microbes, or for rapid identification of bacterial pathogens in drinking water or clinical samples.

## Introduction

Digital Holographic Microscopy (DHM) is a high-throughput tool for imaging microscopic objects in three-dimensional space. It has great utility in biological research as a tool for studying the behaviour of microbes [1–5], and colloidal particles [6, 7]. The three-dimensional swimming behaviour of microbes has been suggested as a way to classify species [8–10], and the simplicity and scalability of the DHM setup allows for its potential use in finding terrestrial or exobiological microbial life [11, 12].

DHM uses the holographic principle to capture 3D information in a 2D image: the interference between light scattered by an object and that from a reference source can be reconstructed to generate a 3D image of the object. The inline configuration uses light from a coherent, monochromatic light source (*e.g.* a laser or LED) passing through a sample chamber where it is scattered by objects of interest. Scattered and unscattered light interfere and the

**Data Availability Statement:** The underlying data has been uploaded to University of York's public repository. (https://doi.org/10.15124/9badfb90-22b9-47be-9e60-d1a492c7e4eb).

**Funding:** LW: EPSRC EP/N014731/1 https://www.ukri.org/councils/epsrc/ SM, JW: Digital Creativity Labs: EPSRC/AHRC/Innovate UK (EP/M023265/1) https://www.ukri.org/councils/epsrc/ VH: The Assuring Autonomy International Programme (http://www.york.ac.uk/assuring-autonomy) The funders played no part in the study design, data collection and analysis, decision to publish, or preparation of the manuscript.

**Competing interests:** The authors have declared that no competing interests exist.

resultant pattern is captured on a detector. The resulting hologram can be used to ascertain 3D information about the sample. This information can be interpreted in a number of ways. A fit to functions obtained from Lorenz-Mie scattering theory can be used to obtain the size, location and refractive index of spheres [13]. If the subject is weakly scattering, the problem reduces to the Rayleigh-Gans condition. This approximation allows objects to be localized in three dimensions using a simple method based on the Gouy phase anomaly [14, 15]. Other methods of analyzing holographic data include heuristic determination of particle volumes based on Rayleigh-Sommerfeld reconstruction. The latter approach has proven suitable for fast visualisation in a virtual reality setting, although it requires considerable computational power [16].

Machine learning (ML) is a method to extract patterns from data and use them to make or help make decisions: example algorithms include logistic regression, naive Bayes or neural networks (NNs) [17]. NNs comprise layers of computational nodes that each take one or more raw input values from the data, apply a non-linear function, and pass the output to the next layer of nodes. Deep learning (DL) neural networks comprise many layers of nodes [17]. These allows a set of learned representations to be built up from the data that are useful for the particular task at hand.

Convolutional neural networks (CNNs) are a type of DL often used with image data. These use a series of convolution kernels combined with sub-sampling 'pooling' layers to build feature maps of the input image [18]. CNNs are trained on datasets to tune internal parameters of the network (the 'weights and biases' of each node) so that the trained network can solve a particular task. Object detection networks utilise CNNs as the basis for finding and classifying objects within images. They do this by locating potential objects of interest within the image and then applying further analysis to that region to find the classification (*e.g.* type of object). Improvements upon this such as the 'you only look once' (YOLO) algorithm family [19] treat object detection as a regression problem to both bounding boxes and object classes.

There are numerous use-cases of DL in various aspects of digital holography and its application in the study of active matter [20, 21]. However, these applications have typically been limited to finding the plane at which a flat object is in focus. CNNs have been used as a classification tool to determine the axial position of Madin-Darby canine kidney cells [22], demonstrating that 'off-the-shelf' DL could be used without any modification or customisation to the task. This has also been treated as a 'regression problem', training the network to give a continuous valued output instead of one of several discrete values. The latter approach is advantageous when the property of interest (e.g. defocus distance) can vary over a wide range, which could otherwise require a lengthy search process [23]. A method to classify two morphologically different bacterial species using holographic images with CNNs has also been presented [24].

CNNs and object detection networks have also been applied to the problem of detecting, localising and analysing holograms of colloidal particles. One approach uses synthetic data to train CNNs to detect areas of interest and then feeds these areas to a second stage of analysis using a fit to Lorenz-Mie scattering theory to obtain the colloid's position, size and refractive index [25]. A single-stage detection method has also been demonstrated, where synthetic images of scattering patterns were used to train a second CNN [26]. The authors in that work report that the localization part of their method attained speeds of about 20ms per frame on idealised synthetic data, with one particle per frame.

One of the main challenges in setting up and training a CNN for object detection is the provision of annotated training data. In the simplest case, images are examined by eye, and annotations made by hand, noting the location and class of objects. This is a laborious and time-consuming process. We overcome this obstacle by using previously verified (but much slower)

DHM analysis routines [4, 14, 27] to provide 'ground truth' annotations for images in a largely automated fashion; these annotated images are then used to train the network offline. By using a deep learning approach, we can approximate the localisation of the cell by training a NN to identify scattering patterns and infer the axial position of the cell from the their size. After the relatively large initial computational cost of training, we have a trained NN that can run this approximation with very few computational resources, increasing the speed of the processing to real-time and running on single-board computers such as the Raspberry Pi. The advantages of being able to perform DHM rapidly, on low powered devices includes the real-time monitoring of fluid flow experiments [28], rapid and remote environmental sampling in terrestrial and extraterrestrial settings [12, 29–32] and detection of infectious cells [33].

Here we present a method of localising cells in three dimensions using YOLOv5 [34], for analysis in tracking routines. We trained the algorithm on a representative database of annotated DHM images of swimming *Escherichia coli* cells that were produced using previously described techniques [4, 27]. We image at relatively low magnification (approximately 1 pixel per micrometer) in order to capture the cells' swimming patterns, which may extend for hundreds of micrometers. At these magnifications, *E. coli* cells (which measure around $1 \times 2\mu m$) can be approximated as point-like scatterers.

## Materials and methods

### Experimental setup

Fig 1 shows a schematic overview of the work flow for training our CNN. Our in-line DHM setup is shown in 'cartoon' form in Fig 1(b). A single-mode fibre-coupled laser diode with peak emission at 642 nm and 15 mW power was mounted in the condenser assembly of a Nikon Ti inverted microscope. This in turn was coupled to a Mikrotron MC-1362 monochrome camera for image acquisition. The objective lens was a ×20 magnification bright field lens with numerical aperture of 0.5. Sequences of 512×512 pixel images (corresponding to a $360 \times 360\mu m$ field of view) were recorded at 30–100 Hz and exposure times of 80–100 μs, for training and analysis. With this setup at this magnification, the cells can be treated as point-like scatterers, so the process is unaffected by the random orientations and variations in morphology of the cell.

### Cell culture and data acquisition

Samples of *E. coli* strain HCB1 were prepared as described elsewhere (*e.g.* [35]) by transferring a crystal of frozen glycerol stock into 10 ml Lysogeny broth (LB). These cultures were grown aerobically overnight to saturation at 31˚C, shaken at 200 rpm. A 20 μl aliquot of the saturated culture was added to 10 ml tryptone broth (TB) and incubated again under the same conditions as before for approximately five hours, to exponential growth phase. A 1 ml aliquot of exponential culture this was centrifuged at 3.3×*g* for 10 minutes to sediment the cells into a pellet, and the supernatant replaced with 1 ml of motility buffer (MB) [35]. The centrifugation and supernatant removal process was repeated twice more to remove residual nutrients. The suspension was diluted with additional motility buffer to a final concentration of approximately 2000 cells/μl. This gave approximately 40 cells per frame in these videos, though some frames contained up to 100 cells. This was found to give a good balance between imaging a suitable number of cells per frame, while ensuring that the single-scattering regime is maintained.

Using this 'dilute' data, a set of 19 videos were recorded: 14 for training, 3 for validation and 2 retained for testing after training was complete. Videos of an empty sample chamber were recorded to provide negative controls for the training. Post processing involved

## Data Acquisition

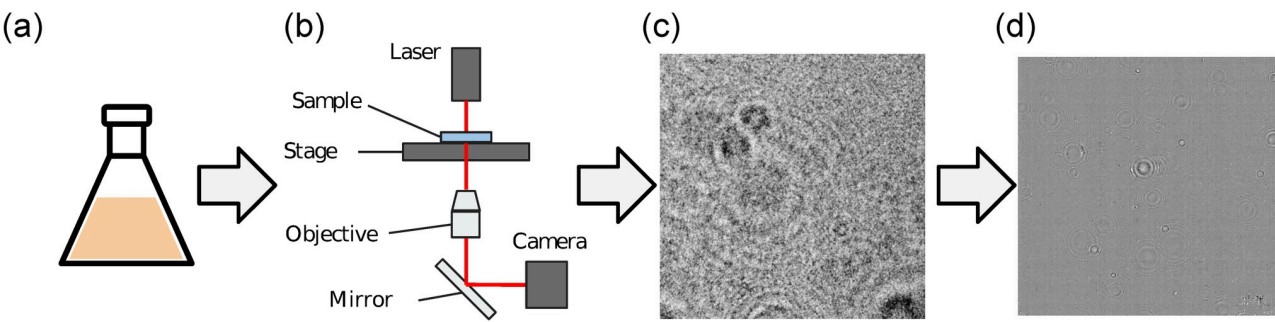

## Data Labelling and Training

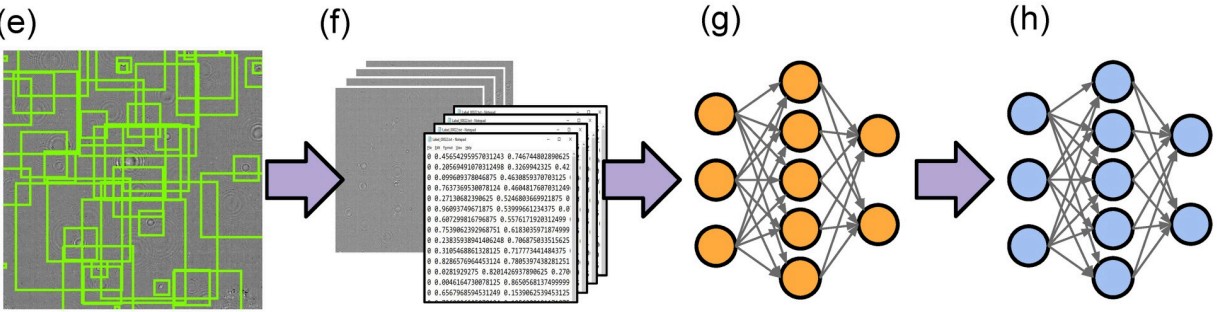

## Inference

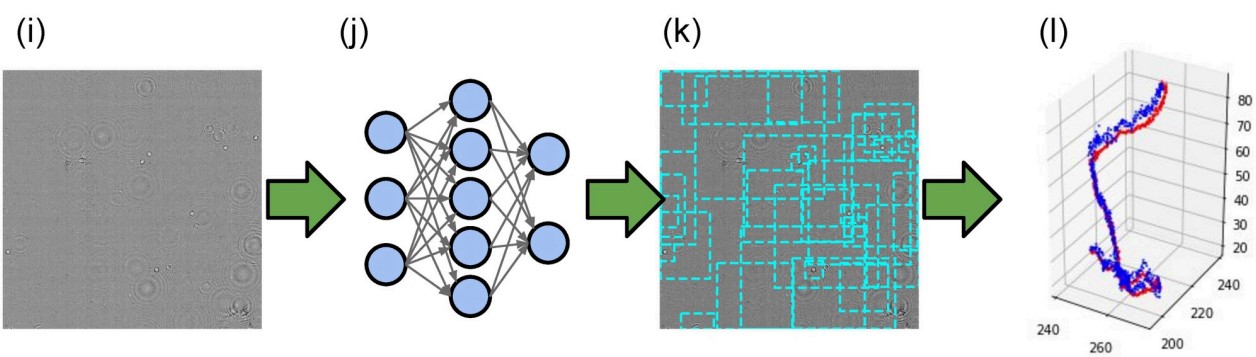

**Fig 1. Workflow of the data collection and training process.** (a) Culturing of *E. coli* as described in the text. (b) 'Cartoon' schematic of the holographic microscope. (c) An example of a raw frame of data, showing scattering from dust on optical elements, diffraction *etc*. The raw video frames are normalized using a median image (see main text) to remove the static background. (d) A normalized image in which the cells can be seen as concentric sets of diffraction rings. (e) DHM-generated Training labels, overlaid on the background-corrected image. (f) Frames and labels are saved in the format required for training a YOLOv5 network. (g) The training was performed on a GPU cluster for 100 epochs (see main text for details). (h) and (j) The trained network. (i) Normalized image of a test video frame. (k) Predicted bounding boxes overlaid on the normalized test image. (l) One of the tracks obtained from the predicted coordinates of the model, compared with the ground truth counterpart.

normalization of the raw video frames in order to improve the contrast of the scattering patterns and remove fixed-pattern noise from optical artifacts. To achieve this, an image was created by taking the median intensity value of each pixel in a video. The pixel values in each frame were then divided by the value at the corresponding location in the median image.

In addition we recorded several videos using samples that were not diluted. This 'undiluted' data contained approximately 240 cells per frame and were used for testing of the effectiveness of the trained model, as shown below.

## Training the neural network

We draw on previously developed holographic microscopy (DHM) methods to obtain 3D cell coordinates from video data, and use this information to train a neural network to perform the same task, orders of magnitude faster. The algorithm that we used requires bounding boxes to be drawn around objects of interest in order to train the network; we automated this task to allow us to greatly expand our quantity of training data. Our previous DHM methods produce a set of 3D coordinates for each video frame. We used the information from these coordinates to identify and draw bounding boxes around cells' images on the raw video frames by hand. Examining around 200 cells in this way, we established a heuristic relationship between bounding box size and a cell's distance from the focal plane (hence the apparent size of its image). Armed with this information, we automated the annotation of $\sim 10^4$ video frames by programatically creating bounding boxes around the tens or hundreds of cells in each video frame. This annotated data was used to train the CNN to automatically output 3D coordinates of cells in each frame, which then required a small polynomial correction to recover the correct cell positions. In the following subsections, we outline the details of each stage.

**Preparation of CNN training data.** We prepared 'ground truth' data using DHM to obtain the three-dimensional positions of objects from raw holographic video data. To achieve this, we used Rayleigh-Sommerfeld back-propagation and the Gouy phase anomaly method to detect objects in an image stack as previously described [14, 36]. Cell tracks were obtained by linking coordinates in neighbouring frames [27]. This approach aids in noise reduction, as spurious, transient signals from within the sample volume are rejected. Cell tracks obtained in this fashion were used to annotate each frame in order to train the YOLOv5 network. The YOLO family of CNN-based algorithms generates bounding boxes around objects in an image; objects that lie further from the focal plane of our microscope produce larger but fainter scattering patterns. We therefore use a heuristic relationship between the size of an object's bounding box, and its axial depth ($z$). The contrast of the object's scattering pattern is related to the refractive index difference between object and background, but the spatial distribution of fringes is (broadly speaking) independent of this in the Rayleigh-Gans scattering regime in which we operate. The apparent extent of the scattering patterns was estimated using the semi-major axis of an ellipse drawn around the pattern, judging by eye where the pattern faded into noise. This allowed us to establish a preliminary relationship between the size of a bounding box and the depth of the object (an approach which we refine below). The defocused diffraction patterns from approximately 200 *E. coli* cells identified by the standard DHM routines were inspected from videos in the training and validation sets. Their apparent radii are plotted against their true axial displacement in Fig 2(a).

The method for determining ground truth tracks finds the positions of cells that are diffusing or swimming. There may be some objects in a given frame from a video that appear superficially similar to a cell scattering pattern by eye, but are not labelled in our training data. It is possible that the trained model will generalise to identify these objects too, but as with the ground truth method, we can use the tracking step to correct for these false positives.

These data show three regimes: axial positions closest to the focal plane (approximately $z \leq 50\ \mu\text{m}$); those in an intermediate range (approximately $50\ \mu\text{m} < z \leq 200\ \mu\text{m}$); and those at greater distances ($z > 200\ \mu\text{m}$). In the intermediate range the relationship between the apparent size of the scattering pattern and its true axial displacement is approximately linear. The

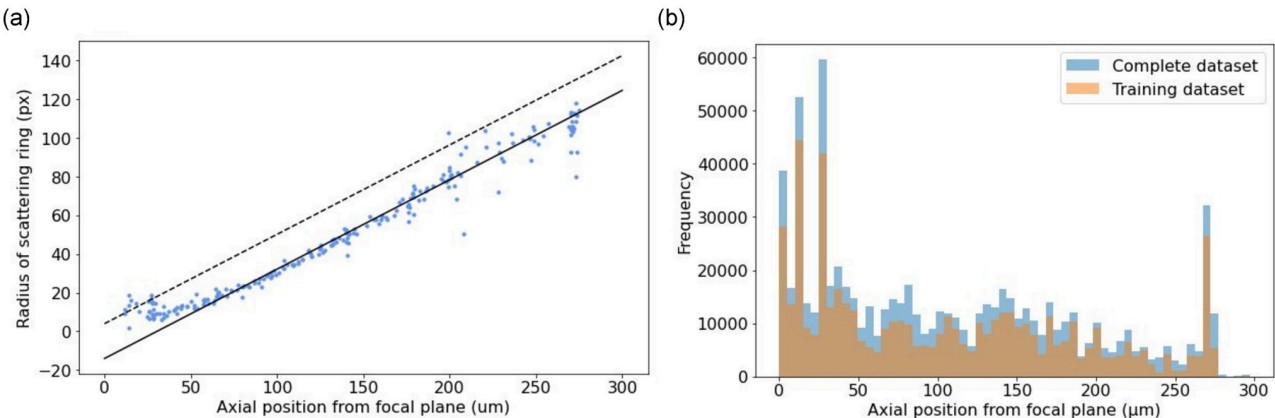

**Fig 2. Data frequency and training fit line.** (a) Establishment of an initial heuristic measure to train a CNN. The apparent extent of the outermost ring of the scattering pattern was measured by hand, and is plotted here against the axial position of the cells. The dashed line is the linear heuristic relationship between the axial position and the bounding box side length used to create the training data for the CNN and is a vertical offset of the solid line (see text). (b) Axial distribution of cells in our sample volume, in both training and test data sets. The data are biased towards smaller axial distances owing to the effects of gravitational sedimentation and the 'wall accumulation' effect in swimming bacteria, observed in previous studies (see text).

solid line in Fig 2(a) is a straight-line fit to the data in this region, but we find that a systematic vertical offset, as indicated by the dashed line, increases the fidelity of reconstruction in later stages. This appears to be due to the inherent uncertainty in estimating the full lateral extent of the scattering pattern by eye (a measure which was estimated conservatively). The offset was added in order to generate positive values for the size of bounding boxes at axial positions closest to the focal plane. The straight line was kept as it was simple, monotonic and was found to work well empirically.

In principle, the physical relationship between the apparent size of the scattering pattern and the cell's axial displacement could be calculated, but in practice this is impeded by the uncertainties in several characteristics that are specific to the apparatus and/or challenging to measure, including the modulation transfer function of the objective lens, the coherence properties of the light, and the noise characteristics of the camera. Effects of some of these properties could be estimated empirically using well-characterised calibration samples (e.g. monodisperse gold nanospheres), but to expedite the application of the method we investigated several heuristic functional forms for the relationship between bounding box size and axial displacement (predominantly polynomials of order 2–4). The training metrics were very similar and the resulting tracks produced by running the network in inference mode were of the same quality or worse, especially in the region $z \leq 50$ μm. We therefore chose a linear model for the relationship between bounding box size and object, for its simplicity. In general, bounding boxes that fit tightly around an object provide better localization information to the model. We speculate that in our case, better results are obtained with over-sized bounding boxes because these are better able to capture poorly-resolved fringes at the outside of the cells' scattering patterns.

From the 14 training and 3 validation videos 14652 training images and 2820 validation images were created; these included 1500 background images in which there were no cells, to act as a negative control. For the CNN training to be effective, we needed to ensure that we have representative images of cells at depths that span our sample chamber (the full extent of which was around 280 μm). The distribution of axial depths of cells in our sample chamber is plotted in Fig 2(b). The cells are motile but subject to gravitational sedimentation, resulting in

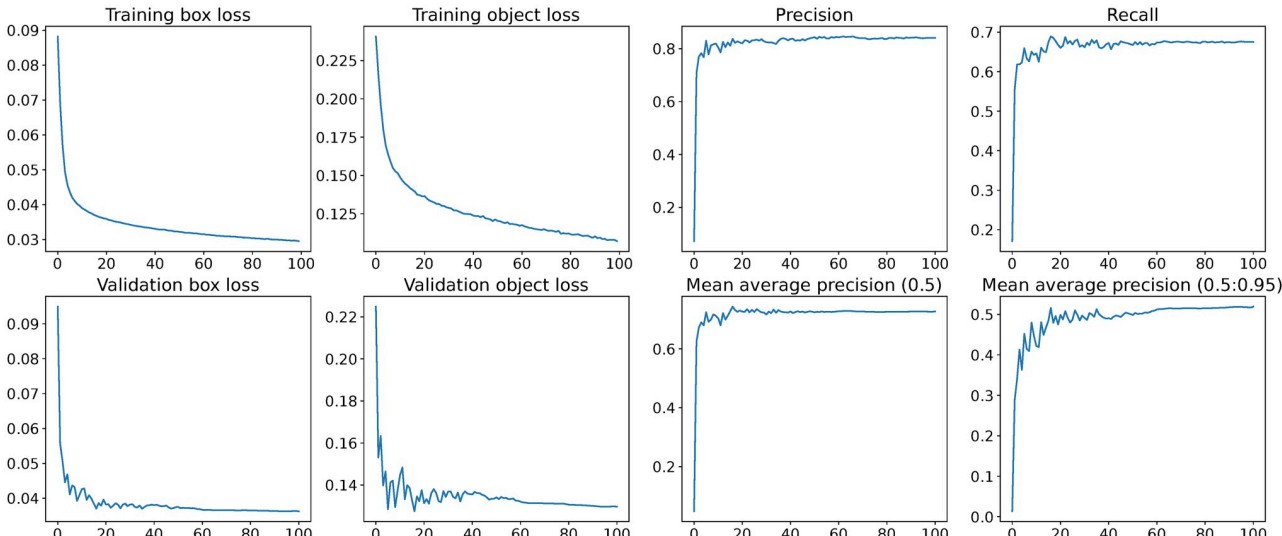

**Fig 3. Training metrics and losses.** Box and object losses for training and validation sets, precision, recall, and mean average precision (mAP) of the network over the 100 training epochs. Box loss is a measure of error between the predicted and ground truth bounding boxes, object loss is a measure of how good the model is at predicting whether an object (a scattering pattern) is present in a bounding box. Precision is the proportion of detected objects that were correct, recall is the ability to detect all objects in an image. A precision recall curve is a useful measure of the trade off between the metrics for a model, and the area under the curve is called the average precision (AP). Intersection over union (IoU) is a measure of the overlap of predicted and ground truth bounding boxes. Mean average precision (mAP) is the mean of the average precision (AP), usually calculated for thresholds of IoU, hence mAP(0.5) is the mAP for objects detected with bounding box IoU of at least 0.5 and mAP(0.5:0.95) is the average of mAP calculated at a range of thresholds from 0.5 to 0.95.

a greater density lower down in the sample chamber. We also note that microorganisms have a tendency to accumulate close to walls [37, 38], resulting in greater density at the boundaries of the chamber (around $z = 0\mu m$ and $z = 270\mu m$). Nevertheless, data in Fig 2(b) show that in the training, validation and test data sets, our data span the full range of available axial positions.

**Network architecture and training.** Several network architectures are available under the YOLOv5 framework. The smallest, 'nano' network (*i.e.* YOLOv5n), was chosen due to its compact architecture and fast inference time on the industry-standard COCO (Common Objects in COntext) dataset. Using 512×512 pixel holographic data, there was no noticeable increase in accuracy when using a larger networks: small (*i.e.* YOLOv5s), medium (*i.e.* YOLOv5m) or large (*i.e.* YOLOv5l). The network was trained for 100 epochs on an Nvidia A40 GPU, with a batch size of 32. To accelerate convergence, weights from the COCO dataset were used for transfer learning [39]. No modification is needed to train the network on greyscale images. Even though these weights are from a model trained on colour images, and used with a 3-channel RGB input, the lower level features that it will have learned, such as light-dark edges, will be universal to object detection and image classification, and will therefore help reduce training time. Results of the training are shown in Fig 3.

Though the 'box loss' and 'object loss' are still decreasing at the point that the training was terminated, the mean average precision (mAP) appears to have saturated, indicating that the weights are likely optimal for generalising to unseen sets [40]. Further training may have continued to decrease the loss values for the training data, but would lead to over-fitting—making the network even better at finding objects in the training images, without increasing performance on unseen images such as those in the validation data. In fact such over-fitting can decrease performance on other data, as the network finds features unique to the training data instead of finding the general features relevant to holographic studies. The resulting network

weights were saved and used in inference mode to detect objects in the (previously 'unseen') test data.

We achieved values for mAP(0.5) of 0.73 and mAP(0.5:0.95) of 0.52. These are an improvement on the benchmarked models, trained on COCO data, of 0.28 and 0.46 respectively [34]. This may be due to the relative simplicity of the setup, as we are only interested in detecting one class of object as opposed to the 80 classes used in the COCO dataset [41]. The maximum possible value of mAP is 1, indicating perfect recall, precision and bounding boxes; mAP is a useful general measure of the relative ability of the model to detect objects. There are several ways to improve performance and achieve higher metrics [34]. The primary way to do this is to use more training data, as the highest performing models can be outperformed by mediocre models trained with much larger datasets [42, 43]. However it is worth noting that improvements are not at all linear with dataset size, so in our case as in others, it may require orders-of-magnitude increases in training set size to gain a few-percentage-point improvement in recognition metrics. Additionally, finding a way to better label the data would help: drawing bounding boxes by hand would probably improve the accuracy somewhat, though this impractical for large training sets.

## Results

We ran the trained CNN in inference mode on 'unseen' videos, the test set from the dilute sample data and videos from the undiluted samples. The network identifies the defocused images of cells (as shown in Fig 1c): the locations of the centres of the diffraction rings shows the $x$, $y$ positions of the cells, and the size of the bounding box is a proxy for their axial position. Examples of predicted bounding boxes and positions for test frames from dilute and undiluted samples are shown in Fig 4. In the case of a rectangular bounding box, such as when the scattering pattern is truncated by the edge of the image, as can be seen in Fig 4(a), the longest edge is used to determine axial position.

(a)                                                      (b)

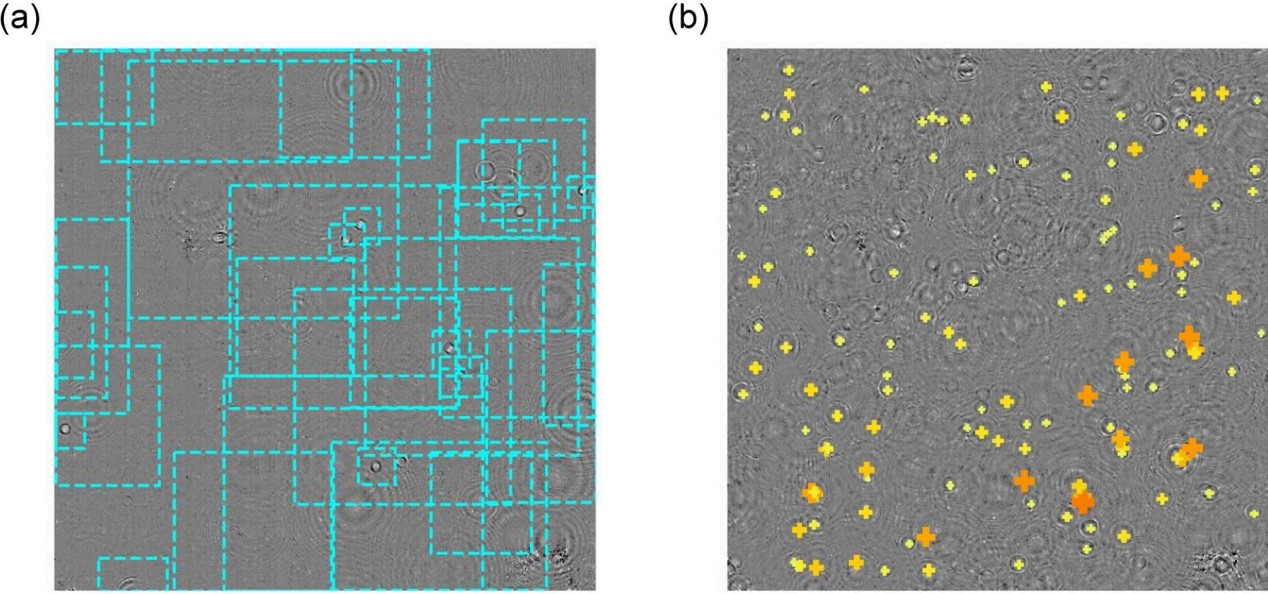

**Fig 4. Cell position and bounding box predictions on unseen video frames.** (a) Dashed cyan lines show the predicted bounding boxes for objects in a frame from the dilute test videos. (b) Crosses show predicted objects in a frame from a more concentrated sample; smaller yellow crosses show small scattering patterns (objects close to the focal plane) and larger orange crosses larger ones (more distant objects).

**Table 1. Metrics for different data sets when run through the trained model in inference.** Descriptions of metrics are found in the figure caption for Fig 3.

| Data Set | Precision | Recall | mAP(0.5) | mAP(0.5:0.95) |
|---|---|---|---|---|
| Training | 0.92 | 0.78 | 0.86 | 0.67 |
| Validation | 0.84 | 0.68 | 0.73 | 0.52 |
| Test | 0.86 | 0.64 | 0.73 | 0.52 |
| Undiluted | 0.45 | 0.41 | 0.40 | 0.28 |

The data from the undiluted sample has too many cells to plot the bounding boxes clearly, so Fig 4(b) shows the predicted location in a frame of undiluted cells with a cross that varies in size and color to indicate the size of the predicted object (and therefore the distance from the focal plane of the cell). We can see in Fig 4(a) that not every scattering pattern has been detected and we know from comparison with the ground truth bounding boxes, generated by the classic DHM algorithms, that some of the objects identified are not scattering patterns produced by cells. Ultimately, the next stage of the process, linking coordinates into tracks (not part of the NN model), remedies these errors.

We evaluated the trained model on videos from the dilute and undiluted samples, and the results are shown in Table 1. Fig 4(b) shows that the model detects plenty of objects in the undiluted sample, but as we can see in Table 1 the metrics for the undiluted samples are lower, failing to detect more of the objects than in the dilute data. This is likely due to the overlapping scattering patterns and rings making the image very noisy, and making it harder to accurately detect the objects above the confidence threshold. As a result the inferred coordinates are sparser and slightly noisier, and generating tracks from the coordinates is harder. Using data from undiluted samples to train the model may improve this. Promisingly, we note that the speed of detection using the CNN is approximately constant regardless of the number of cells in the image.

Fig 5(a) shows the results of operating our trained CNN on unseen data. The axial positions of cells reported by the CNN are plotted against those obtained by DHM for both validation and test data. The coordinates obtained by the CNN show a systematic deviation from the 'true' (DHM) coordinates in both data sets, increasing with depth. To correct this deviation, we apply a 4$^{th}$-order polynomial correction to the axial positions reported by the CNN. We note that this correction is qualitatively different in nature and purpose to that presented in Fig 2(a): the latter was used as an 'initial guess' in training the CNN to construct bounding boxes around cell scattering patterns. The black dashed line in Fig 5(a) shows a target relationship between the CNN and DHM data, and the red dot-dashed line represents the 4$^{th}$-order polynomial fitted to the validation data only. Fig 5(b) shows the test data after the correction has been applied. Although we initially suspected that the choice of heuristic in the previous section was the source of this systematic error, we found that the need for this correction polynomial was not eliminated by using a higher order fit for the axial position-bounding box size heuristic. It appears that the CNN generalises to produce bounding boxes that are a smaller fit to the detected objects than expected, but in a systematic way that allows for automatic correction. The data inferred by the trained YOLOv5n network, suitably corrected, are in good agreement with the ground truth (Fig 5c) and certainly sufficient to estimate parameters of biophysical interest such as swimming speed, and time between reorientation events. Using the root mean squared error (RMSE) between CNN and ground truth tracks to quantify this difference, we find that the RMSE for uncorrected tracks is 3.01 μm and corrected tracks is 1.99 μm. The performance of the trained CNN deteriorates when the cell lies at axial positions $z > 150$ μm from the focal plane. We believe this is due to the decreased signal-to-noise ratio

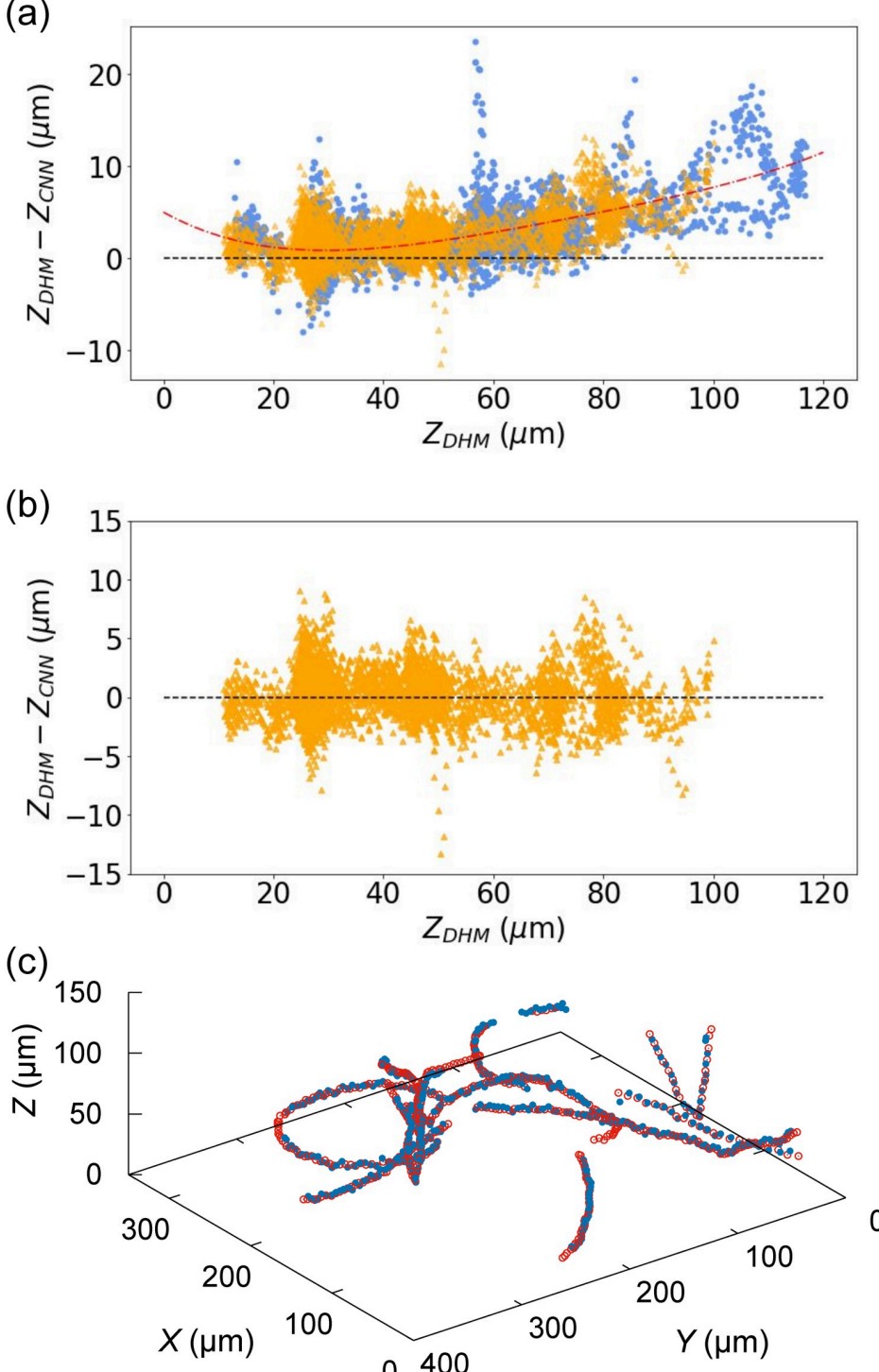

**Fig 5. Results of inference on the validation and test videos.** Data from test videos are represented by orange triangles, and data from validation videos are plotted with blue circles. (a) Raw axial coordinates obtained by the trained YOLOv5 as a function of 'true' axial depth established by DHM. The red dot-dashed line is a fourth-order fit to the validation data. (b) Test video axial coordinates, as obtained in panel (a), after removal of fourth-order correction factor (see text). (c) Selected tracks obtained by three-dimensional, AI-enabled tracking after the second round of correction (blue solid points), plotted with the original tracks obtained by DHM (red empty points).

**Table 2. Inference time per frame using different processors.**

| Processor | Time per frame (ms) |
|---|---|
| Nvidia 3060 GPU | 10.5 |
| Nvidia Jetson Nano on GPU | 74.6 |
| Raspberry Pi 4B (8GB RAM) | 373.5 |
| Desktop CPU | 149 |
| Standard DHM on Desktop CPU (4-core Intel Xeon, 3.07 GHz CPU and 12GB RAM (98 GFLOPS)) | 1658 |

for objects lying further from the focal plane, as their images are larger but fainter, making the application of a bounding box more challenging. Nevertheless, 3D tracking in a sample depth of 120 μm is more than sufficient for many biophysical studies.

To benchmark the performance of the trained YOLOv5n network in the context of holographic microscopy, the trained network was tested against 'unseen' videos on an Nvidia 3060 GPU (12GB GDDR6 VRAM, 3584 CUDA cores, 12.74 TFLOPS), an Nvidia Jetson Nano single board computer (4GB LPDDR4 RAM, 128 CUDA cores, 235.8 GFLOPS), and a Raspberry Pi 4B (8GB SDRAM, 1.8GHz Quad core Cortex-A72 processor, 24 GFLOPS). A median image was used to remove the static background before the images were processed; in the future this median could be generated from the first few frames of a live video before inference begins. The inference speeds using these test configurations are shown in Table 2; these are the speeds for the CNN to produce the coordinates. We achieved a performance of > 13 frames per second on a GPU-accelerated single-board computer like the Nvidia Jetson Nano, opening the door to real-time feedback in experiments studying the stimulus response of planktonic cells, performed by lightweight computational hardware. Although the tracks obtained from the CNN are slightly noisier than their DHM equivalents, for many applications this will be an acceptable trade-off for the ability to run at real-time speeds.

The image formation process in a microscope depends on the light source and lenses used, and the system's performance is often characterised by a modulation transfer function (MTF). Each optical system will have a slightly different MTF, and so the images of microscopic objects will be subtly different in different microscopes. We do not anticipate that this specificity extends to the identity of an individual lens or image sensor, but a change in (for example) illumination wavelength or objective lens numerical aperture will change the optimal network weights, and we would anticipate degraded performance if our trained network were to be applied unmodified to a different optical system. This sensitivity of object detection networks to small perturbations in the input data has been studied in other settings. For example, they are known to be susceptible to slight alterations in images, such as subtle changes to color, that would not affect a human's ability to identify an object in an image [44]. Transferring our method to a new setup would require some retraining using images or sequences generated by the new optical setup, to adapt the network weights to the new environment. Fortunately, this process is made more efficient through transfer learning. In the same way that we made use of the pre-trained COCO dataset weights, a network trained on our setup could be adapted to achieve the same accuracy on a different optical setup using far fewer training examples and less training time than we required [39].

Given the rod-shaped nature of *E. coli* cells and natural variation in shape and size, there is some variation in the scattering rings due to the cells' random orientations in the plane. This can be seen in Fig 4, with some scattering patterns showing slight deviation from the circular

patterns where the long axis of the body is in the plane. Across the large number of cells imaged in our training data, we expect that we sample the full distribution of cell orientation and size from within the standard population from a lab-grown bacterial culture. Clearly our trained model is fully capable of identifying these in the test videos, as we were able to reconstruct multiple tracks that traverse paths encompassing all possible orientations, as shown in Fig 5(c). At higher magnifications, the effect of variations in cell image resulting from variable cell morphology and orientation would be more pronounced, leading to a potential breakdown in the relationship between bounding box size and depth shown in Fig 2(a). However, with a large enough, representative training set we anticipate that this method also would work at these magnifications, albeit with potential trade-offs in axial range and noise.

In fact, our method could be used to identify cell morphology at higher magnifications, or to identify cells with morphology different to that of coliform cells, such as spirochaetes. This could be achieved by creating a training dataset representative of enough different cell morphologies, treating the axial depth as a separate target value unrelated to boundary box size. This would likely require an enormous dataset, and lies outside the scope of the current study, but we briefly discuss it here as an example of how our process could be extended. The problem of recognising and categorising microbes based on shape and location might be best treated as a classification task, with the cells' axial depth binned into classes (*e.g.* 1, 5, or 10 μm bins). For instance, the class of cells that are all 95 μm$< z \leq 100$ μm will contain a variety of scattering patterns of subtly differing shapes, depending on the cell's geometry. This information is contained within the scattering pattern, and therefore it is theoretically possible to train our CNN using this data. The higher the desired axial resolution, the more classes would be required and the more training data would be needed to provide sufficient examples for each class. Our unmodified method is still suitable for use at magnifications where objects can be treated as point-like scatterers, and the resulting tracks could be used to determine if the objects are motile or undergoing Brownian motion—an indicator of life in the case of exobiological life detection.

## Conclusion

We have demonstrated that with suitable calibration, the size of bounding boxes in an object detection network can be used as a fast and accurate proxy for the axial depth of cells in holographic video microscopy. This was achieved by assuming a simple relationship between axial depth and the detectable radius of the scattering patterns in DHM images. We find that the substantial reduction in computing complexity when using a CNN provides two key, interlinked benefits: the ability to process data two orders of magnitude faster than traditional DHM 3D-tracking methods, or to process data in real-time on modest hardware, such as single-board computers like the Nvidia Jetson Nano. The latter will allow the technique to be used in challenging experimental situations, such as *in situ* data collection in hazardous environments. In particular, our implementation is suited to situations where limited processing power and bandwidth constraints make the storage or transmission of uncompressed video impossible. More broadly, it is interesting to note that a CNN developed to identify macroscopic objects, possibly with occlusion, can be repurposed to function in a situation where the scattering patterns of objects overlap in an additive fashion, as seen in Fig 1(d). Doubtless, specialist algorithms could improve the performance of image recognition in DHM, but the fact that an off-the-shelf algorithm can be repurposed to this application shows the promise for improvements in power and speed that artificial intelligence methods can bring to digital holography.

## Acknowledgments

We would like to thank Steve Smith for helpful discussions.

## Author Contributions

**Conceptualization:** Samuel A. Matthews, Carlos Coelho, Victoria J. Hodge, James A. Walker, Laurence G. Wilson.

**Data curation:** Samuel A. Matthews, Carlos Coelho, Erick E. Rodriguez Salas, Emma E. Brock.

**Formal analysis:** Samuel A. Matthews.

**Funding acquisition:** Laurence G. Wilson.

**Investigation:** Samuel A. Matthews, Erick E. Rodriguez Salas.

**Methodology:** Samuel A. Matthews, Carlos Coelho, Erick E. Rodriguez Salas, Victoria J. Hodge, James A. Walker, Laurence G. Wilson.

**Project administration:** Victoria J. Hodge, James A. Walker, Laurence G. Wilson.

**Software:** Samuel A. Matthews, Erick E. Rodriguez Salas.

**Supervision:** Victoria J. Hodge, James A. Walker, Laurence G. Wilson.

**Validation:** Samuel A. Matthews.

**Visualization:** Samuel A. Matthews.

**Writing – original draft:** Samuel A. Matthews.

**Writing – review & editing:** Samuel A. Matthews, Victoria J. Hodge, James A. Walker, Laurence G. Wilson.

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
