## [Decision Letter · Decision Letter 0]

10 Jan 2024

PONE-D-23-42223Real-time 3D tracking of swimming microbes using digital holographic microscopy and deep learningPLOS ONE

Dear Dr. Wilson,

Thank you for submitting your manuscript to PLOS ONE. After careful consideration, we feel that it has merit but does not fully meet PLOS ONE’s publication criteria as it currently stands. Therefore, we invite you to submit a revised version of the manuscript that addresses the points raised during the review process.

We look forward to receiving your revised manuscript.

Kind regards,

Jay Nadeau, PhD

Academic Editor

PLOS ONE

Journal Requirements:

3. In the online submission form, you indicated that your data will be submitted to a repository upon acceptance.  We strongly recommend all authors deposit their data before acceptance, as the process can be lengthy and hold up publication timelines. Please note that, though access restrictions are acceptable now, your entire minimal  dataset will need to be made freely accessible if your manuscript is accepted for publication. This policy applies to all data except where public deposition would breach compliance with the protocol approved by your research ethics board. If you are unable to adhere to our open data policy, please kindly revise your statement to explain your reasoning and we will seek the editor's input on an exemption.

Reviewers' comments:

Reviewer's Responses to Questions

**Comments to the Author**

1. Is the manuscript technically sound, and do the data support the conclusions?

Reviewer #1: Yes

Reviewer #2: Yes

2. Has the statistical analysis been performed appropriately and rigorously? 

Reviewer #1: I Don't Know

Reviewer #2: Yes

3. Have the authors made all data underlying the findings in their manuscript fully available?

Reviewer #1: No

Reviewer #2: Yes

4. Is the manuscript presented in an intelligible fashion and written in standard English?

Reviewer #1: Yes

Reviewer #2: Yes

5. Review Comments to the Author

Reviewer #1: This article presents an analysis of integrating Convolutional Neural Networks (CNN) with Digital Holographic Microscopy (DHM), demonstrating its potential utility, particularly when prior knowledge of the dataset is applied. The method employs a straightforward strategy of correlating the bounding box size with the Z-plane, a practical solution for scenarios where computational resources are limited and detailed reconstructions are unnecessary.

However, the article would benefit from addressing the following additional aspects:

An exploration of the method's robustness across different cell morphologies within the same species. It's crucial to ascertain if cells with varied morphologies yield consistent results and how the technique fares with cells oriented in different planes.

An assessment of the technique's efficacy with denser samples.

A detailed discussion on the specificity and robustness of the method concerning the system's variables, such as the modulation transfer function of the objective lens, the coherence properties of the light, and the camera's noise characteristics.

Moreover:

Is this approach 10 times faster than the conventional method when talking about a larger order? If so, does this include the time of the reconstruction process with the other method?

A schematic representation of the method, including the preparation and training phases, would be highly beneficial for comprehension.

In Fig 4c, it's unclear whether these are selected tracks. Additionally, the maximum value for mAP appears to end at 0.5; further explanation and an analysis of Fig 3's weaknesses would enhance understanding.

Suggestions for optimizing the training to attain higher mAP values would be valuable.

The article also needs clarification regarding the number of cells per image as mentioned in lines 104 and 121. Additionally, there are several typographical errors that need rectification:

Line 118: "this"

Line 169: "cite"

Line 247: "?"

Line 253: Missing "."

No data available yet?

Reviewer #2: Dear authors, I have read your manuscript in detail, and I gladly recommend it for publication with revisions. Please carefully consider my comments below.

Summary

This manuscript demonstrates the acceleration of 3D particle tracking in digital holographic microscopy video in a two-step process. First, a YOLOv5 CNN object detection model identifies bounding boxes around particles’ Airy rings in median-subtracted holograms. Second, the sizes of these bounding boxes are related to the axial position of the particle. This process is faster and cheaper computationally compared to existing methods, enabling real-time DHM analysis with common hardware.

Major items

Major 1, Fig 2, L155: The figure, its caption, and the text do not clearly communicate why and how the systematic vertical offset was applied. The figure seems to represent an incorrect straight-line fit to the data shown at first glance. An improvement would be to include both pre- and post-offset fits. The text should describe how the vertical offset was calculated; the slope and intercept of the fit itself could also be reported, considering its simplicity. It is also unclear whether this offset is strictly necessary, since the 4th-order polynomial correction applied to the model post hoc could also compensate for this error (unless the argument is that the model itself is simply worse without the offset). However, since re-training the deep learning model requires significant effort, a clearer explanation here is sufficient.

Major 2, L79-82: While this particular instrument’s configuration allows the microbes to be treated as point-like scatterers, a discussion of the method’s limitations with high-magnification instruments where the size of the microbe itself contributes to the bounding box size is critical (e.g. a large microbe at the focal plane could have the same bounding box size as a small microbe at some z). This would be especially challenging in an environment with a diverse population of microbes with different morphologies. While less relevant for applications such as environmental monitoring, instruments for exobiological life detection may require high magnifications for such investigation.

Major 3, L204: The results section of this manuscript does not report any quantitative metrics for tracking performance. While the manuscript may not include other methods to compare to, a quantitative performance metric is important for objective evaluation and for future works seeking to improve upon these results on the released data set. An advanced option would be the alpha and beta metrics used for the Particle Tracking Challenge (Chenouard, et al., 2014, Nature Methods), but a simple RMSE would also be sufficient. Additionally, performance metrics on the training, validation, and test sets would also indicate or dispel overfitting.

Major 4, L217: The 4th-order polynomial correction (which could also be described as a form of model calibration) should be fit on the results from the training data, then applied unchanged to the results from the unseen data. By fitting the correction directly on the unseen data, it hides the possibility that this bias/deviation introduced by the model cannot be corrected in a generalizable fashion (as suggested in L229). This is a critical correction that should require little effort.

Minor items

L68: I disagree with “we can think of the trained NN as approximating the reconstruction and localisation processes.” Per my Major 2 comment, if this were the case, this method would be resilient to microbes with different sizes. Instead, the trained NN is simply faster and more accurate at measuring the sizes of Airy rings, which has a direct relationship with the axial position and size of particles. Since the size is assumed to be fixed, the position can be retrieved.

Figure 1d: When printed in grayscale, the red bounding boxes disappear against the gray background. Use a brighter or lighter color such as green or cyan for print accessibility.

L118: Typo, “thsi”

L143: Describe the efforts taken to standardize this hand-estimation for reproducibility; for example, the bounding boxes likely needed to be drawn around a consistent number of Airy rings. Additionally, specifying the “scattering pattern” also as Airy rings may be clearer to readers more familiar with DHM techniques.

L158: While theoretically calculating this relationship could be challenging, could it also be empirically measured more reliably with calibration beads?

L169: Citation marked as missing, but a citation here is unnecessary. Simply, bounding boxes fit tightly around an object provides better localization information to the model.

L186: “Digital Footprint” is an existing term that refers to a user’s traceable digital activities on the internet https://en.wikipedia.org/wiki/Digital_footprint . “compact architecture,” “fewer parameters,” or “lower complexity” would be more appropriate.

L189: Specifying the larger model that was attempted would be informative for reproduction and future work, if available.

L191: A pretrained model on COCO would expect 3-channel RGB images, whereas DHM images are grayscale. Specify how this adaptation was made. A comparison of whether transfer learning substantively improved model performance would also be valuable, but not required.

L192: A reader unfamiliar with deep learning for object detection would not understand the significance of ‘box loss’ and ‘object loss’, and ‘precision’ and ‘recall’ as presented in Figure 3 is also ambiguous in terms of object detection (bounding boxes above a certain IoU threshold?) Include some additional description as to what these loss and metric values represent.

L195: “(here the validation images)” seems to be an incomplete thought.

L207: “used by the code” is jarring; the phrase can be deleted.

L215: YOLO does not always produce square bounding boxes - how is the “radius of the scattering ring” (per Figure 2) derived for rectangular bounding boxes?

L232: Typo, “percormance”

L241: Reporting the theoretical 32-bit floating point (FP32) FLOPS of the hardware would be more informative of each computer’s performance compared to memory and the number of CUDA cores. This is sometimes reported by the manufacturer, or can be computed from the clock speed and the number of cores.

Figure 4a: The blue dashed line is very difficult to see against the purple points, even on a monitor. Please use better-contrasting colors for accessibility.

L247: There is a LaTeX error resulting in an upside down question mark.

L252: “to many users” may be replaced with “for many applications.”

L253: Typo, missing period at the end of the sentence.

Table 1: Adding a column of frames per second may be helpful for the reader.

L260: “ODN” here is confusing, as it is only used once at L36, and it is not a common acronym in the field (as evidenced by a Google search). Continue using “CNN”, or specify “object detection CNN.”

L261: I am not certain what “traditional (back-propagation style) DHM” refers to. If it refers to the previous DHM tracking method cited as [4, 29, 30], it should simply refer to “traditional DHM 3D-tracking methods.” The following “or” should also be an “and.”

L264: “in situ” is latin and has no hyphen.

L7, L73, L265: These recent references may be relevant for motivating the need for DHM 3D tracking methods under compute and bandwidth constraints, and are worth considering.

The 2023 Planetary Science Decadal Survey (p.376) http://doi.org/10.17226/26522, which calls out the importance of real-time analysis and autonomy for extraterrestrial life detection.

Dubay, et al., 2023 https://doi.org/10.1016/j.mimet.2022.106658, which surveys 2D and 3D DHM tracking methods, and describes the computational complexity of such methods.

Wronkiewicz, et al., 2023 (preprint) https://arxiv.org/abs/2304.13189, which implements a 2D particle tracker on raw DHM images for exobiology detection on spaceflight processors.

6. PLOS authors have the option to publish the peer review history of their article (what does this mean?). If published, this will include your full peer review and any attached files.

Reviewer #1: No

Reviewer #2: No

---

## [Author Response · Author response to Decision Letter 0]

26 Feb 2024

Please see attached document detailing responses to reviews, and the marked up version of our resubmission that indicates where the document has been amended.

---

## [Editor Report · Decision Letter 1]

13 Mar 2024

Real-time 3D tracking of swimming microbes using digital holographic microscopy and deep learning

PONE-D-23-42223R1

Dear Dr. Wilson,

We’re pleased to inform you that your manuscript has been judged scientifically suitable for publication and will be formally accepted for publication once it meets all outstanding technical requirements.

Kind regards,

Jay Nadeau, PhD

Academic Editor

PLOS ONE